# Identification and Mining of Functional Components of Polyphenols in Fruits of *Malus* Germplasm Resources Based on Multivariate Analysis

**DOI:** 10.3390/foods13213465

**Published:** 2024-10-29

**Authors:** Dajiang Wang, Guangyi Wang, Xiang Lu, Zhao Liu, Simiao Sun, Hanxin Guo, Wen Tian, Zichen Li, Lin Wang, Lianwen Li, Yuan Gao, Kun Wang

**Affiliations:** 1Research Institute of Pomology, Chinese Academy of Agricultural Sciences (CAAS), Key Laboratory of Horticulture Crops Germplasm Resources Utilization, Ministry of Agriculture and Rural Affairs, Xingcheng 125100, China; wangdajiang@caas.cn (D.W.); wanggy0315@163.com (G.W.); xianglu1997@163.com (X.L.); lz__0427@163.com (Z.L.); sunsimiao@caas.cn (S.S.); 82101235244@caas.cn (H.G.); tianwen8025@163.com (W.T.); lishencheno@163.com (Z.L.); juziwanglin@163.com (L.W.); lilianwen@caas.cn (L.L.); gaoyuan02@caas.cn (Y.G.); 2Agricultural College of Shihezi University, Xinjiang Production and Construction Corps Key Laboratory of Special Fruits and Vegetables Cultivation Physiology and Germplasm Resources Utilization, Shihezi 832003, China

**Keywords:** *Malus sieversii*, fruits, polyphenol, peel, pulp, multivariate analysis, LC–MS

## Abstract

Polyphenols are important functional components that have anti-cancer and anti-inflammatory effects. Apple fruit is rich in polyphenols and is one of the dietary sources of polyphenols. The polyphenol components and contents of the peel and pulp of 74 *Malus sieversii* (Led.) Roem. and 26 Chinese *Malus* germplasm resources were determined using ultra-high-phase chromatography (UPLC) and liquid chromatography–mass spectrometry (LC–MS). The results showed that 34 components were detected in the peel and 30 in the flesh, and that the polyphenol components and contents of the different germplasm resources were significantly different; the polyphenol content of *Malus sieversii* (Led.) Roem. was significantly higher than that of the other local varieties, and the polyphenol content in the peel was also higher than that in the flesh. Rutin, quercetin 3-O-arabopyranoside, kaempferol 3-O-rutinoside, and peonidin 3-O-galactoside were detected only in the peel. The total polyphenol content in the peel ranged from 949.76 to 5840.06 mg/kg, and the polyphenol content in the pulp ranged from 367.31 to 5123.10 mg/kg. The cluster analysis of polyphenol components and contents in peel and pulp showed that 100 *Malus* germplasm resources could be grouped into four categories. Principal component analysis of 34 kinds and 30 kinds of polyphenols in peel and pulp of 100 resources was performed. If the eigenvalue is greater than 1, eight and seven principal components are extracted, respectively. Five *Malus* resources with high polyphenol content in the peel and pulp were selected: ‘XY-77’ (peel: 5840.06 mg/kg, pulp: 5123.10 mg/kg; ‘LF-09’ (peel: 4692.63 mg/kg, pulp: 3729.79 mg/kg); ‘2012-5’ (peel: 4377.61 mg/kg, pulp: 3847.54 mg/kg); ‘29028’ (peel: 5088.05 mg/kg, pulp: 3994.61 mg/kg); and ‘11-01’ (peel: 5154.45 mg/kg, pulp: 3616.15 mg/kg). These results provide us with information regarding the polyphenol composition and content of the wild apple resources and local cultivars. The high polyphenol content resources obtained by screening can be used as raw materials for the extraction of polyphenol components and functional fruit juice processing and can also be used as parents for functional fruit creation and variety breeding.

## 1. Introduction

The apple is a deciduous tree of *Malus* in the Rosaceae family and one of the four largest sources of fruit in the world [1]. There are about 80 countries and regions in the world engaged in apple production [2]. Apple cultivation has a long history in China and was called ‘Nai’ in ancient times. It was first recorded in ‘Shanglin Fu’ by Simaxiangru and was also recorded during later dynasties; thus, it has a written history that is more than 2000 years old [3].

Due to the origin and evolution of the world’s apple plants and the different ecological and geographical environments, five gene centers have formed, including those of North America, Central Asia, Europe, the Caucasus, Eastern China, and East Asia. Among the five major gene centers, China, the center of origin in East Asia, has the most species of *Malus* plants, and it is the center of the genetic diversity of the world’s *Malus* plants. There are about 35 species of *Malus* in the world, including 6 cultivated species and 21 wild species originating from China [4]. In 1959, Zhang estimated that the area of *Malus* sieversii (Led.) Roem. forest in Yili Valley was about 12,000 ha, with 84 types, which not only had all the fine qualities of modern cultivated apples but also showed more abundant diversity in terms of fruit types, color, quality, and other agronomic traits [5,6]. In addition to being an important gene pool of fruit trees in China, it is the most important original gene pool for the breeding and genetic improvement of cultivated apples. It is also the ancestor species of cultivated apples [7,8,9,10].

In recent years, with the pursuit of health and the increasing demands of people for a better life, more and more attention is being paid to food safety and the nutritional value of food, and fruit has become an important source of dietary supplements [11,12]. As the saying goes, “An apple a day keeps the doctor away”, and apples have gained considerable attention in research studies around the world due to their rich content of various nutrients, such as polyphenols and bioactive substances. The apple is the third-highest dietary source of polyphenols after tea and onions. Apple polyphenols have significant antioxidant capacity and play an important role in preventing food deterioration and improving food quality and shelf life. Apple polyphenols also have the following healthcare functions: they have positive antibacterial and anti-inflammatory effects; they prevent obesity, diabetes, and cardiovascular diseases; and they have anti-tumor and anti-aging properties [13,14,15,16,17].

The polyphenol components and contents of apples vary greatly according to variety, tissue, growing environment, and maturity [18]. The polyphenol content of wild apple resources is usually higher than that of the cultivated varieties. The polyphenol content of wine apples is generally higher than that of fresh apples and juice apples, and the polyphenol content of the bitter apple fruit is higher than that of sour apples and sweet apples [19,20,21]. As the ancestor of the cultivated apple, the polyphenol component and content characteristics of *Malus sieversii* (Led.) Roem. fruit must be studied. It has important theoretical and practical significance for the breeding of functional varieties of apples. In this study, 74 *Malus sieversii* (Led.) Roem. and 26 local varieties grown in the “National Repository of Apple Germplasm Resources (Xingcheng)” were used as test materials to determine the polyphenol components and contents of the fruit peel and pulp. The differences in polyphenol content between *Malus sieversii* (Led.) Roem. and the local varieties were studied. Cluster analysis and PCA analysis were performed to find the difference in polyphenol components and content among the 100 *Malus* germplasm resources, identify and mine functional components of polyphenols in fruits of *Malus* germplasm resources based on multivariate analysis, and screen the *Malus* germplasm resources with high polyphenol content. It will provide material data for the functional variety creation and breeding of apples.

## 2. Materials and Methods

### 2.1. Test Material and Sample

All of the 100 *Malus* germplasm resources used as experimental materials were collected from the “National Repository of Apple Germplasm Resources (Xingcheng)”, Institute of Pomology, Chinese Academy of Agricultural Sciences (Appendix A). The nursery land management level was medium, and the rootstock was *Malus baccata* (L.) Borkh.; the sizes were relatively equal. A total of 30–50 mature fruits of uniform size and free from disease, insect, and mechanical damage were collected from the periphery of the tree crown, with 10–15 fruits per replicate; in particular, 3 biological replicates were used. After harvest, the fruits were transported back to the laboratory at room temperature; then, the core was removed, the peel was sliced, the pulp was cut into pieces, and the fruit was frozen in liquid nitrogen and stored in the refrigerator at −80 °C for testing.

### 2.2. Method

The polyphenol extraction method followed the methods of Nie et al. and Wang et al. [12,22,23,24], with slight modifications. The peel and pulp of the fruit were frozen and ground to a powder weighing 5.0 g (fresh weight, FW); the powder was placed in a 50 mL centrifuge tube (BD Falcon^®^, Corning, New York, NY, USA) with 25 mL of 80% ethanol, shaken well, and placed away from light for 12 h; it was then subjected to ultrasonic treatment (SB 25-12 DTD, Ningbo, China). After 20 min of vibration, it was centrifuged at 10,000 r/min for 5 min (CF 16 RX, Hitachi, Japan), and the supernatant was absorbed. The residue was added to 20 mL of 80% ethanol for repeated extraction; the supernatant was combined twice, and the volume of 80% ethanol was fixed at 50 mL. The ethanol was removed by the evaporation of 10 mL of the extract using a 40 °C rotary evaporator (R-215, Buchi, Switzerland). The OasisHLB solid phase extraction column (Oasis^®^HLB, Waters, MA, USA) was activated with 10 mL of methanol and 10 mL of purified water. The solid phase extraction column was washed twice with 5 mL of deionized water, and the waste liquid was discarded. The solid phase extraction column with 5 mL of methanol was washed twice, and the filtrate was collected. The filtrate collected by the rotary evaporator was evaporated at 40 °C to nearly dry; the methanol volume was fixed at 5 mL, and the filtrate was filtered through a 0.22 μm nylon (Nylon66) organic phase filter membrane (Jinteng, Tianjin, China) into a brown vial for measurement.

Polyphenols were detected using UPLC-XeVo/TQ ultra-high-performance liquid chromatograph (UPLC-XEVO /TQ) with PDA eλ detector (Waters, MA, USA), and LC-10 ATvp high-performance liquid chromatograph (Shimadzu, Japan). The columns were ACQUIITY UPLC^®^ HSS T3 1.8 μm and XSelect^®^ HSS T3 5 μm from Waters Corporation.

Uplc-xevo /TQ ultra-high-performance liquid chromatograph, ultra-performance liquid chromatography (UPLC) conditions: the flow rate was 0.3 mL/min; the sample size was 2.0 μL; the column temperature was 40 °C; the wavelength scanning range was 200–600 nm; the quantitative detection wavelength was 280 nm (flavanol), 320 nm (hydroxycinnamic acid), and 360 nm (flavonol); the mobile phase A was 0.5% formic acid solution, and the mobile phase B was acetonitrile. Gradient dewashing, liquid B, 0% (0 min) → 10% (1 min) → 20% (10 min) → 25% (16 min) → 40% (18 min) → 100% (19 min), 20 min back to the initial state, balance for 5 min.

Dihydrochalcone polyphenols should be detected separately under UPLC conditions. The UPLC conditions were as follows: flow rate of 0.3 mL/min; sample size of 2.0 μL; column temperature of 40 °C; quantitative detection wavelength of 280 nm; mobile phase A as 0.5% formic acid solution; mobile phase B as acetonitrile. Gradient dewashing, B solution, 0% (0 min) → 8% (2 min) → 15% (10 min) → 23% (20 min) → 100% (20.5 min) → 100% (21.5 min), 22 min return to the initial state, balance for 5 min.

The conditions for liquid chromatography–mass spectrometry (LC–MS) are as follows: in the electrospray ionization (ESI) and multiple reaction monitoring (MRM) modes, the ion source temperature is 150 °C; the desolvent temperature is 450 °C; and the desolvent gas flow rate is 650 L/h. The flow rate of cone-hole gas is 50 L/h, and the flow rate of collision gas (high-purity argon) is 0.14 mL/min.

LC-10 ATvp High-Performance Liquid Chromatography (High-Performance Liquid Chromatography, HPLC) conditions are as follows: the flow rate was 0.7 mL/min; the sample size was 1 μL; the column temperature was 40 °C; and the wavelength was quantitatively detected at 520 nm (anthocyanins). The mobile phase A is 5% formic acid, and B is a 1:1 mixture of formic acid and acetonitrile. Gradient dewashing, solution B: 5% (0 min) → 10% (10 min) → 20% (30 min) → 30% (40 min) → 90% (40.5 min) → 90% (44.5 min) → 5% (50 min), return to the initial state, balance for 20 min.

The polyphenolic compositions without standards were identified using LC–MS and quantified using UPLC. Polyphenol standards of catechin, chlorogenic acid, epicatechin, rutin, quercetin 3-O-xyloside, cyanidin 3-O-galactoside, cyanidin 3-O-glucoside, cyanidin 3-O-arabinoside, peonidin 3-O-galactoside, cyanidin 3-O-xyloside, and phlorizin were purchased from Sigma-Aldrich (St. Louis, MO, USA); procyanidin B1, procyanidin B2, procyanidin C1, quercetin 3-O-galactoside, quercetin 3-O-glucoside, quercetin 3-O-arabinoside, and quercetin3-O-rhamnoside were purchased from ChromaDex (Irvine, CA, USA). 4-*p*-coumarylquinic acid and 5-*p*-coumarylquinic acid were quantified by the chlorogenic acid standard. 3-hydroxyphloretin-xylglucoside, 3-hydroxyphloretin-glucoside, phloridin-hexose-hexose, and phloretin xyloglucoside were quantified by phloridzin [24].

### 2.3. Data Analysis

Microsoft Office Excel 2021 and SPSS 27 (SPSS Inc., USA) were used to conduct a single-factor ANOVA test of the data, and the significance, mean value, standard deviation, and proportion were calculated. SPSS 27 and Origin 2021 were used for principal component analysis (PCA). Hiplot (https://hiplot.com.cn/home/index.html, accessed on 2 September 2024) and ChiPlot (https://www.chiplot.online/, accessed on 2 September 2024) were used for hierarchical clustering analysis; the components and contents of polyphenols in peel and pulp of 100 fruits were used as variables; the Ward D2 method and Euclidean distance were used for cluster analysis [25]. The contents of five types of polyphenols were calculated as the sum of the contents of the various polyphenol components, and the total polyphenolic content was calculated as the sum of the contents of all the polyphenol components.

## 3. Results

### 3.1. Fruit Polyphenol Components of Malus Germplasm Resources

Through liquid chromatography–mass spectrometry analysis, a total of 34 kinds of polyphenol components were detected in the apple fruits, including five flavanols and seven dihydrochalones at a wavelength of 280 nm (Figure 1A), as follows: procyanidin B1 (PROB1); catechin (CATE); procyanidin B2 (PROB2); epicatechin (EPI); procyanidin C1 (PROC1); 3-Hydroxyphloretin 2’-xylglucoside (HYDXY); phloretin hexose-hexose (PHLHE); 3-Hydroxyphloretin 2’-glucoside (HYDGL); phloretin xyloglucoside (PHLXY); phloretin pentose-hexose (PHLPE); phloretin pentose-hexose (isomers, PHLPE1); and phlorizine (PHLZI).

Three hydroxy-cinnamic acids were detected at a wavelength of 320 nm (Figure 1B): chlorogenic acid (CHLAC); 4-*p*-coumaroylquinic acid (4COUAC); and 5-*p*-coumaroylquinic acid (5COUAC).

Thirteen flavonol polyphenol substances were detected at a wavelength of 360 nm (Figure 1C): rutin (RUTIN); quercetin 3-O-galactoside (QUEGA); quercetin 3-O-glucoside (QUEGL); quercetin 3-O-xyloside (QUEXY); quercetin 3-O-arabinopyranoside (QUEPY); kaempferol 3-O-galacoside (KAEGA); kaempferol 3-O-rhanosylglucoside (KAERG); quercetin 3-O-arabinofuranoside (QUEFU); kaempferol 3-O-gluctoside (KAEGL); quercetin 3-O-rhamnoside (QUERH); Kaempferol 3-O-arabinopyranoside (KAEPY); kaempferol 3-O-arabinofuranoside (KAEFU); and kaempferol 3-O-rhamnoside (KAERH).

Six anthocyanins were detected at 520 nm wavelengths (Figure 1D): cyanidin 3-O-galactoside (CYAGA); cyanidin 3-O-glucoside (CYAGL); cyanidin 3-O-arabinoside (CYA3AR); peonidin 3-O-galactoside (PEOGA); cyanidin 7-O-arabinoside (CYA7AR); and cyanidin 3-O-xyloside (CYAXY).

### 3.2. Differences in Polyphenol Components and Contents of Fruits of Malus Germplasm Resources

Of the 100 *Malus* germplasm resources, 76 were wild resources, and 24 were local varieties. The average total polyphenol content in the peel and pulp of the wild resources was 2842.28 mg/kg and 1809.63 mg/kg, respectively, and that of the local varieties was 1744.36 mg/kg and 1058.25 mg/kg, respectively. The total polyphenol content in both the peel and pulp of the wild resources was higher than that of the local varieties (Table 1).

The average total polyphenol content in the peel and pulp of the 100 resources was 2578.78 mg/kg and 1629.30 mg/kg, respectively, and the overall total polyphenol content in the peel was higher than that in the pulp (Figure 2A). The total polyphenol content in the peel and pulp of ‘XY-77’ was the highest at 5840.06 mg/kg and 5123.10 mg/kg, respectively. The lowest total polyphenol contents were 640.14 mg/kg in the peel of ‘Hongxun 2’ and 367.31 mg/kg in the pulp of ‘HQ-3’.

The polyphenol components and contents of the peel and pulp of 100 fruits were used as variables; cluster analysis was performed on 100 *Malus* germplasm resources. As shown in Figure 2B, the 100 resources were copolymerized into four categories, and the resources ‘XY-77’, ‘2012-5’, ‘LF-09’, ‘11-01’, and ‘29028’, with high total polyphenol contents in the peel and pulp, were grouped into the first category. Most of the local varieties from Hebei and Xinjiang were grouped together with the resources of low total polyphenol content. The remaining wild resources were grouped into the second group according to the difference in the total polyphenol content between the fruit peel and pulp.

The contents of five kinds of polyphenols in the peel and pulp of 100 apple germplasm resources were significantly different. In the peel, flavonols were the dominant polyphenols of the fruit, with an average content of 1657.72 mg/kg, accounting for 61.28% of the total polyphenol content. The flavanol content of ‘XY-77’ was the highest, reaching 4503.58 mg/kg and accounting for 77.12% of the total polyphenol content; the flavanol content of ‘EM-3’ was the highest, reaching 80.68% with a content of 3121.06 mg/kg. Flavonol was the second most abundant polyphenolic substance in the peel, with an average content of 309.49 mg/kg, accounting for 25.78%. The ‘29016’ had the highest content of flavonol, reaching 874.00 mg/kg and accounting for 23.13% of the total polyphenolic content; ‘HC-4’ had the highest content of flavonol, accounting for 23.13% of the total polyphenolic content. It reached 35.09%, and the content was 407.05 mg/kg. The average content of dihydrochalcone was 400.94 mg/kg, accounting for 15.49%. The content of dihydrochalcone in ‘HZ-16’ was the highest, reaching 1004.22 mg/kg and accounting for 13.52% of the total polyphenol content; the content of dihydrochalcone in ‘Balenghaitang’ was the highest, reaching 30.73% with a content of 368.37 mg/kg. The proportion of hydroxycinnamic acid in the total polyphenol content in the peel was relatively low; the average content was 205.31 mg/kg, accounting for 9.34%. The hydroxycinnamic acid content of ‘EM1-6’ was 800.63 mg/kg, accounting for 33.78%, which was the highest. Anthocyanins were detected in the red apple peel, and ‘Hongxun 2’ had the highest content and proportion of anthocyanins, reaching 100.31 mg/kg and 15.67%, respectively (Figure 3).

In the pulp, flavanols were also dominant, with an average content of 1188.61 mg/kg, accounting for 69.30%. The flavanol content of ‘XY-77’ was the highest, reaching 4061.37 mg/kg and accounting for 79.28% of the total polyphenol content, and the flavanol content of ‘HC-52’ was the highest, accounting for 94.71% and reaching 2130.78 mg/kg. Hydroxycinnamic acid was the second most abundant polyphenolic substance in the pulp, with an average content of 361.85 mg/kg, accounting for 25.54%. The hydroxycinnamic acid content of ‘EM1-6’ was the highest (1050.42 mg/kg, accounting for 57.29%), and the content of hydroxycinnamic acid in ‘Hongxun2’ was the highest (82.38%, accounting for 420.45 mg/kg). The content of dihydrochaldone in the pulp was low; the average content was 75.11 mg/kg, and the average proportion was 4.87%. The highest content and the proportion of dihydrochalcone in ‘29016’ were 208.88 mg/kg and 13.25%, respectively. Flavonols were rarely detected in the pulp, with an average content of 3.42 mg/kg and an average proportion of 0.22%. The flavonol content of ‘HC-52’ was the highest (18.64 mg/kg), accounting for 0.83%; the flavonol proportion of ‘HK-5’ was the highest (1.32%), accounting for 8.87 mg/kg. Anthocyanins were only detected in ‘Hongxun 2’ in the pulp; the content was 31.05 mg/kg, accounting for 6.08% (Figure 3).

The average contents of the different polyphenol components in the 100 resources were analyzed, and the average content and proportion of each polyphenol component are shown in Table 2. Thirty-four components were detected in the peel and thirty in the pulp. Rutin, quercetin 3-O-arabopyranoside, kaempferol 3-O-rutinoside, and paeoniflorin 3-O-galactoside were detected only in the peel. Proanthocyanidin B2, epicatechin, and proanthocyanidin C1 were dominant in the peel and pulp of the fruit. The average contents and the proportions of proanthocyanidin B2 were 607.24 mg/kg and 23.55% in the pericarp and 456.95 mg/kg and 28.05% in the pulp. The average contents of epicatechin and proanthocyanidin C1 were above 16%. Chlorogenic acid was dominant only in the pulp, with an average content and proportion of 311.32 mg/kg and 19.11%. The content in the peel was significantly lower than that in the pulp, but it was also an important polyphenol substance, with an average content and proportion of 178.92 mg/kg and 6.94%. The average contents of phloridin-xyloglucose and phloridin in the fruit peel were also higher, with average contents and proportions of 205.84 mg/kg and 7.98%, 164.93 mg/kg, and 6.40%, respectively. Among other polyphenol components, the average contents of nine components in the peel were above 1%; these were proanthocyanidin B1, catechin, quercetin 3-O-galactoside, quercetin 3-O-glucoside, quercetin 3-O-xyloside, quercetin 3-o-arabinofuranoside, and quercetin 3-o-rhamnoside. The average content of 4-*p*-coumaroylquinic acid in the pulp was above 1%. Although the average content and proportion of centaurin 3-O-galactoside were low, it was the most important component of the anthocyanins.

### 3.3. Principal Component Analysis of Polyphenols in Peel and Pulp of Apple Germplasm Resources

Principal component analysis of 34 kinds of polyphenols in the peel of 100 *Malus* resources was carried out. According to the eigenvalue of greater than 1, a total of eight principal components were extracted, and the cumulative variance contribution rate was 82.94%. The contribution rate of principal component 1 (PC1) was 24.54%; that of principal component 2 (PC2) was 16.16%, and that of principal component 3 (PC3) was 13.90%. The cumulative contribution rate of the first three principal components was 54.60% (Appendix A). According to the composition matrix of the different principal components (Appendix A), the characteristic components of PC 1 included quercetin 3-O-glucoside, kaempferol 3-o-arabinofuranoside, rutin and kaempferol 3-o-glucoside, and other components. The characteristic components of PC 2 included quercetin 3-O-arabinopyranoside, quercetin 3-O-galactoside, epicatechin, quercetin 3-O-xyloside, and other components. The main characteristic component of PC3 was anthocyanins. The characteristic components of principal component 4 (PC4) included catechins, proanthocyanidins B1, and 4-*p*-coumaroylquinic acid. The main characteristic component of principal component 5 (PC5) was 3-hydroxy-phloridin-xyloglucose. The characteristic components of principal component 6 (PC6) were chlorogenic acid and 5-*p*-coumaroylquinic acid. The main characteristic component of principal component 7 (PC7) was paeoniflorin 3-O-galactoside. The characteristic component of principal component 8 (PC8) was 3-hydroxy-phloridin-glucose.

In the peel, six *Malus* resources had strong specificity. The higher anthocyanin content of ‘Hongxun 2’ was different from that of the other resources. The contents of chlorogenic acid and 5-*p*-coumaroylquinic acid in ‘EM1-6’ were different from those of the other resources. The content of kaempferol in ‘LF-24’ was different from that of the other resources. The ‘29016’, ‘HDM-19’, and ‘26007’ were significantly different from the other resources, with high levels of rutin and quercetin 3-O-glucoside (Figure 4).

Principal component analysis of 30 kinds of polyphenol components in the pulp of 100 *Malus* resources was carried out. According to the eigenvalue of greater than 1, a total of seven principal components were extracted; the contribution rate of PC 1 was 24.25%, the contribution rate of PC 2 was 17.83%, and the contribution rate of PC 3 was 14.66%. The cumulative contribution rate of the first three principal components was 56.74%, and when the number of principal components was seven, the cumulative variance contribution rate was 84.96% (Appendix A). According to the composition matrix of the different principal components (Appendix A), the main characteristic components of PC 1 included procyanidin C1, phloridin-pentose-hexose 1, proanthocyanidin B2, and epicatechin. PC 2 was kaempferols. In PC 3, the main characteristic component was anthocyanins. PC 4 included phloretin-pentose-hexose, quercetin 3-O-arabinofuranoside, quercetin 3-O-xyloside, and phloretin-xyloglucose. The main characteristic components of PC 5 included chlorogenic acid, quercetin-galactoside, catechin, 5-*p*-coumaroylquinic acid, and proanthocyanidin B1. PC 6 was divided into phloridin, 3-hydroxy-phloridin-glucose, and 4-*p*-coumaroylquinic acid. The characteristic component of PC 7 was quercetin 3-O-rhamnoside.

In the pulp, there were four *Malus* resources with strong specificity. The higher anthocyanin content of ‘Hongxun 2’ was different from that of the other resources. The content of kaempferol in ‘EM3-2’ was different from that of the other resources. The higher contents of proocyanidin C1, phloridin-pentose-hexose 1, proanthocyanidin B2, and epicatechin in ‘LF-09’ and ‘XY-77’ were significantly different from those of the other resources (Figure 5).

### 3.4. Cluster Analysis of Polyphenols in Peel and Pulp of Malus Germplasm Resources

The polyphenol contents in the peel and pulp of 100 apple germplasm resources were ranked from high to low. Using the quartile method, the top 1–25% of the resources were classified as grade A, 26–50% as grade B, 51–75% as grade C, and 76–100% as grade D. Subsequently, the contents of 34 kinds of polyphenol components in the peel and 30 kinds of polyphenol components in the pulp were standardized, and hierarchical cluster analysis was carried out. 

As shown in Figure 6A, the 34 polyphenol components in the peel were grouped into five categories, which were significantly different from the classification according to chemical structure. Chlorogenic acid and 5-*p*-coumarylquinic acid were grouped together, and the contents of chlorogenic acid and 5-*p*-coumarylquinic acid in the grade A and B resources were mostly lower than those in the grade C and D resources. Kaempferol 3-O-rhamnoside, kaempferol 3-O-galactoside, kaempferol 3-O-glucoside, kaempferol 3-O-rutinoside, kaempferol 3-O-arabopyranoside, kaempferol 3-O-pyranoside, 3-hydroxy-phloridin-xyloglucose, 4-*p*-coumaroylquinic acid, and the anthocyanin compounds were grouped into one group, and the content level was weakly correlated with the total polyphenol. However, it was an important standard component of the resource-clustering classification. Phloridin-hexose-hexose, phloridin-xyloglucose, phloridin-pentose-hexose, phloridin-pentose-hexose 1, proanthocyanidin B2, epicatechin, and proanthocyanidin C1 were clustered into one group, and their contents in the grade A and B resources were generally higher than those in the grade C and D resources. Among them, proanthocyanidin B2, epicatechin, and proanthocyanidin C1 were the dominant components of the apple polyphenols, and the content level determined the content of the total polyphenols. 3-hydroxy-phloridin-glucose, phloridin, proanthocyanidins B1, and catechins were grouped into one group. Rutin, quercetin 3-O-galactoside, quercetin 3-O-glucoside, quercetin 3-O-xyloside, quercetin 3-O-arabinoside, quercetin 3-O-arabinoside, quercetin 3-O-arabinoside, quercetin 3-O-rhamnoside, and kaempferol 3-O-arabinofuranoside, were grouped into one group.

The 100 *Malus* resources of four grades were grouped into five types, with the polyphenol components and contents in the peel as variables. The level A, B, and C resources were more dispersed, and the level D resources were more concentrated. The class D resources were mainly clustered in one class, and the low flavonol content was the main basis for the clustering of this class of resources. The content of anthocyanins was an important basis for the clustering. ‘Hongxun 2’, ‘HK-9’, and ‘HC-52’ were clustered with a high content of anthocyanins and low contents of hydroxycinnamic acid and kaempferol. Kaempferol was also an important clustering index, and higher kaempferol content was usually accompanied by higher quercetin content. The ‘29016’, ‘LF-19’, ‘XY-43’, ‘26007’, ‘the HDM-23’, ‘the HDM-29’, ‘the EM1-4’, ‘LF-24’, ‘HHD-8’, ‘the HDM-19’ and ‘EM-3-2’ were clustered together in one category. The ‘HF-04’, ‘HK-4’, ‘HZ-7’, ‘HX-11’, ‘EM1-6’, and ‘HZ-16’ were clustered with high contents of chlorogenic acid, 5-*p*-coumaroylquinic acid, and quercetin. The remaining group was clustered with high levels of proanthocyanidins B2, epicatechin, and proanthocyanidins C1 and low levels of quercetin (Figure 6A).

As shown in Figure 6B, the 30 polyphenol components in the pulp were grouped into four categories. Phloridin-xyloglucose, phloridin-pentose-hexose, phloridin-pentose-hexose 1, proanthocyanidin B2, epicatechin, and proanthocyanidin C1 were clustered into one group. The contents of these seven polyphenol components in the grade A and B resources were generally higher than those in the grade C and D resources. 3-hydroxy-phloridin-glucose and phloridin were grouped together. 5-*p*-coumaroylquinic acid, quercetin-galactoside, quercetin-glucoside, kaempferol, and anthocyanins were clustered into one group. These components were not detected in the pulp of most resources, which was an important basis for the clustering analysis. The other components were grouped into one category, which had a weak correlation with the total polyphenol content. The higher content of these components in the different resources usually represented the higher total polyphenol content in the resource pulp, but the content levels of these components in the resources with a higher total polyphenol content in the pulp were significantly different.

Among the 100 *Malus* resources, the grade A and B resources were mainly divided into two types, one with a high content of each component and the other with high contents of seven components, namely, phloridin-xyloglucose, phloridin-pentose-hexose, chlorogenic acid, phloridin-pentose-hexose 1, proanthocyanidin B2, epicatechin, and proanthocyanidin C1, and low contents of the other components. The grade C and D resources were more concentrated, and the content of these seven components was also the main classification standard. The medium content level was mainly in the grade C resources, and the low content level was mainly in the grade D resources. According to the polyphenol components and the contents of the pulp, 100 *Malus* resources were grouped into six categories, among which four categories and five resources had strong specificity. “Hongxun 2” was a class of anthocyanins detected in the pulp, and “EM3-2” was a class of kaempferol substances. The contents of polyphenols in ‘XY-77’ were higher, except for phloridin and 4-*p*-coumaroylquinic acid. ‘EM1-6’ and ‘HK-4’ were clustered with quercetin glucoside, the chlorogenic acid content, and the phloridin xyloglucose content (Figure 6B).

## 4. Discussion

The polyphenol content of the different apple types was significantly different, and the polyphenol content of the wild resources was usually higher than that of the local varieties and cultivated varieties; this was mainly related to apple domestication [26]. Improving fruit taste was one of the most important goals of apple breeding. Studies have shown that the polyphenol content of the main cultivated apple varieties was significantly different from that of the traditional apple varieties, and the traditional varieties usually had higher polyphenol content [27]. The content of polyphenols was mainly related to the astringency and bitter taste of fruits, and the removal of astringency and bitter taste during breeding was an important reason for the decrease in polyphenol content [28,29]. The composition and content of apple polyphenols were also affected by the origin of the apple. Guo et al. [30] studied the polyphenol diversity of 145 apple germplasm resources from more than 10 countries and found that varieties from different regions had significant effects on both the total polyphenol and single polyphenol components. Wang et al. [12] studied the polyphenols of 103 *Malus* germplasm resources from different origins and found that North China was densely populated. The apples were greatly affected by artificial selection, and the contents of polyphenol components that affected the astringency and bitterness of the fruits were significantly reduced, resulting in the total polyphenol content in North China being generally lower than that in Northwest China. Following many years of study, Chen et al. found that the wild apple in Xinjiang had high flavonoid content, rich aroma, and great potential for further exploitation [31]. In this study, the analysis of the total polyphenol content of 100 *Malus* germplasm resources reached the same conclusion: the average total polyphenol content of the wild resources was higher than that of the local varieties, and the average total polyphenol content of the resources from Xinjiang was higher than that of Hebei. *Malus sieversii* (Led.) Roem. had greater utilization potential in functional peel development.

The content of the polyphenol components in different parts of the apple fruit was significantly different [32]. In this study, the total polyphenol content in the peel was generally higher than that in the pulp; this finding was consistent with those of previous studies [33,34]. The apple polyphenols were mainly divided into five categories: flavanols; dihydrochalcones; hydroxycinnamic acid; flavonols; and anthocyanins [35]. Flavanols were the main polyphenols of apple polyphenols, with the highest contents in both the peel and the pulp [36]. Alonso et al. [37] studied the polyphenol composition characteristics of 31 Basque apples and found that the average content of flavanols in the peel, pulp, and juice was between 40% and 80%, while the average contents of flavanols in the peel and pulp in this study were 62.21% and 69.78%, respectively. In the pulp, hydroxycinnamic acid was also the main polyphenol substance. Khanizadeh et al. [38] studied the polyphenol composition of eight Canadian processed apple varieties and found that the content of hydroxycinnamic acid in the pulp could reach 30% of the total polyphenol, while the content in the peel was only 10%. In this study, the proportion of hydroxycinnamic acid in the apple resources was also similar and accounted for 9.34% in the peel and 25.54% in the pulp.

A total of 34 kinds of polyphenols were detected in 100 *Malus* germplasm resources, 30 of which were detected in the fruit pulp. Rutin, quercetin 3-O-arabopyranoside, kaempferol 3-O-rutinoside, and paeoniflorin 3-O-galactoside were detected only in the peel. It was generally believed that quercetin substances mainly existed in the peel and that the pulp did not contain rutin, quercetin 3-O-galactoside, and other substances [39,40,41,42]. However, studies have also proven that small amounts of these substances were detected in the pulp. LamperiI et al. detected quercetin 3-O-galactoposide and quercetin 3-O-glucoside in the pulp of ‘Annurca’, ‘Golden Delicious’, ‘Red Chief’, and ‘Stayman Neepling’ apples [43]. In this study, quercetin 3-O-galactoside was detected in the pulp of only two resources; quercetin 3-O-glucoside was detected in the pulp of five resources, and kaempferols were detected only in ‘EM-3-2’; these findings were basically consistent with those of previous studies [44]. The differences in the polyphenol components between the peel and pulp are still controversial. With the development and optimization of detection technology, the components and contents of the polyphenols detected in apples of the same variety are increasing. Studies have shown that the polyphenol content detected using liquid chromatography–mass spectrometry is 4~18% higher than that detected using light spectrophotometry in the same material [45]. Wu Qian analyzed the polyphenol substances of the “Hongmantang” apple fruit using metabolomics, and a total of 210 kinds of polyphenol substances were detected, but several quercetin derivatives were not detected in its pulp [46]. It can be inferred that the components and diversity of polyphenols in the apple fruits were also affected by the detection methods.

In the principal component analysis, the dispersion degree among the different resources was high, and the correlation between the total polyphenol content and polyphenol components was weak. The difference in the peel contents of the different resources was significantly related to the diversity of the resources [30]. In this study, the difference in the peel contents of the different resources was mainly reflected in the content of anthocyanins and flavonols, which may be related to the polyphenol content, which is easily affected by the environment. Light was an important factor affecting the synthesis and metabolism of the polyphenols in the fruit peel, and some studies have shown that anthocyanins and flavonols were more sensitive to light. In a study on the effect of light on the polyphenol content in “Fuji” apples, Jakopic et al. found that the apples at the top of the tree crown contained higher levels of anthocyanins and flavonols than those in the apple peel inside the tree [47]. Similar conclusions were also obtained regarding the effects of bagging on the polyphenol content in apple fruits. Bagging reduced the content of most polyphenol components in the peel and pulp, and the degree of polyphenol content decrease in the peel was significantly higher than that in the pulp [48,49].

Epicatechin, proanthocyanidins C1, and proanthocyanidins B2 were the most important polyphenols of the apple polyphenols, usually accounting for more than 50% of the total polyphenols, and they directly affected the level of the total polyphenol content [50,51]. In this study, it was also found that phloridin-pentose hexos-1 and the total polyphenol content had the same change trend. The average contents of phloridin-pentose hexosaccharide 1 in the apple polyphenols were 6.66 mg/kg in the peel, accounting for 0.26% of the total polyphenol content, and 5.05 mg/kg in the pulp, accounting for only 0.3% of the total polyphenol content. However, in the principal component analysis and cluster analysis of the peel and pulp, phloridin-pentose hexos-1 showed a similar change trend to that of epicatechin, proanthocyanidin C1, and proanthocyanidin B2, the three main polyphenol components. Studies have shown that phloretin and its derivatives are almost exclusively found in apples and can be said to be apple-specific substances [52]. Phloretin has been used in the chemical classification and differentiation of roseaceae plant species and to identify key substances in apple juice and other fruit juices [53,54]. At the same time, the content of phloridins is less affected by environmental factors such as light [46]. It can be inferred that phloridin-pentose-hexose 1 has the potential to be a key substance for the evaluation of the polyphenol content in apples and may be an important part of the evaluation criteria for determining the polyphenol content in apples.

Through principal component analysis and cluster analysis of the peel and pulp, seven specific resources were selected. Among them, ‘Hongxun 2’ had higher anthocyanin content in the fruit peel and pulp, which was significantly different from the other resources. In apples, anthocyanins existed only in the red peel and pulp, and the color of the peel and pulp was related to the accumulation of anthocyanins [55]. Anthocyanins also had various physiological functions, and red flesh apples had stronger antioxidant capacity and anti-proliferation activity than white flesh apples [56]. Anthocyanins had important functions in terms of both appearance quality and nutritional quality, and the high-polyphenol apple resources rich in anthocyanins had broad application prospects [57]. ‘EM-3-2’ had a variety of kaempferol substances detected in the peel and pulp. Kaempferol had important physiological functions in biomedicine and played an important role in relieving asthma [58]. ‘XY-77’, ‘LF-09’, ‘2012-5’, ‘29028’, and ‘11-01’ were all resources with a high polyphenol content. Among them, ‘XY-77’ was the resource with the highest comprehensive total polyphenol content; the total polyphenol content of the fruit peel reached 5840.06 mg/kg, and the total polyphenol content of the fruit pulp reached 5123.10 mg/kg. The level of total polyphenol content affected various biological activities, such as antioxidant capacity [46,59]. The antibacterial, anti-inflammatory, and anticancer abilities of chlorogenic acid could play an important role in suppressing gingivitis, preventing oral cancer, and treating oral diseases [60]. Methyl mercaptan was the main component of oral odor, and apple polyphenolic compounds could inhibit the production of methyl mercaptan, showing a significant dose effect [61]. Phloridin and proanthocyanidins could make melanin fade and whiten skin [62]. Therefore, the resources with high total polyphenol and special polyphenol component content were of great value to the breeding of functional varieties.

## 5. Conclusions

The polyphenol components and contents in the peel and pulp of 100 *Malus* germplasm resources were identified. Thirty-four polyphenol components were identified in the peel, and 30 polyphenol components were identified in the pulp. The content of polyphenols in the peel of the *Malus* germplasm resources was higher than that in the pulp, and the content of polyphenols in the wild resources was higher than that in the local varieties. One hundred *Malus* germplasm resources could be grouped into four categories by cluster analysis, and eight and seven principal components are extracted in peel and pulp, respectively, by PCA analysis. Five *Malus* germplasm resources with high polyphenol content were selected from *Malus sieversii* (Led.) Roem., which was the ancestor of the cultivated apple and possessed all the traits of the modern cultivated apple. Therefore, in *Malus sieversii* (Led.) Roem., in particular, five high polyphenol resources were screened that could provide raw materials for apple functional fruit development and should be the focus of functional fruit research in the future. In addition, future studies can study the dynamic change in polyphenol content in the fruit development process of the high polyphenol resources obtained by screening and explore the period of the highest content of polyphenol components for substance extraction and utilization.

## Figures and Tables

**Figure 1 foods-13-03465-f001:**
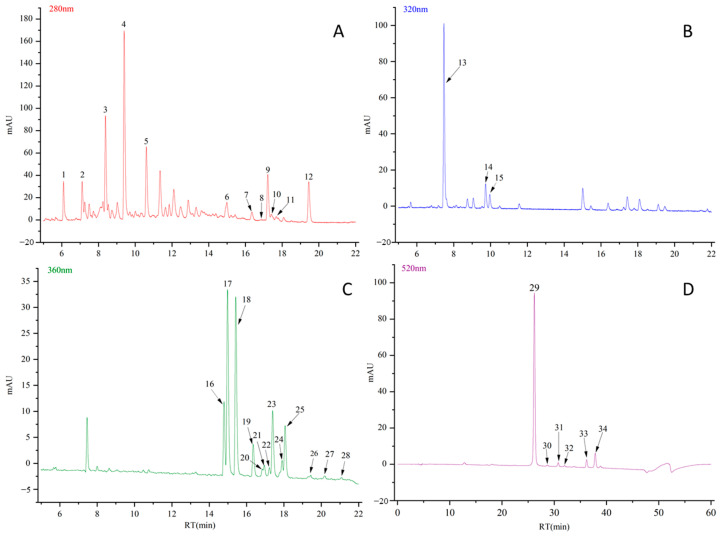
Liquid chromatography of 34 polyphenolic components detected in the samples. (**A**) Chromatogram at 280 nm absorption peak. (**B**) Chromatogram at 320 nm absorption peak. (**C**) Chromatogram at 360 nm absorption peak. (**D**) Chromatogram at 520 nm absorption peak. Note: 1: procyanidin B1; 2: catechin; 3: procyanidin B2; 4: epicatechin; 5: procyanidin C1; 6: 3-hydroxyphloretin 2’-xylglucoside; 7: phloretin hexose-hexose; 8: 3-hydroxyphloretin 2′-glucoside; 9: phloretin xyloglucoside; 10: phloretin pentose-hexose; 11: phloretin pentose-hexose1 (Isomers); 12: phlorizine; 13: chlorogenic acid; 14: 4-*p*-coumaroylquinic acid; 15: 5-*p*-coumaroylquinic acid; 16: rutin; 17: quercetin 3-O-galactoside; 18: quercetin 3-O-glucoside; 19: quercetin 3-O-xyloside; 20: quercetin 3-O-arabinopyranosidearabinopyranoside; 21: kaempferol 3-O-galacoside; 22: kaempferol 3-O-rhanosylglucoside; 23: quercetin 3-O-arabinofuranoside; 24: kaempferol 3-O-gluctoside; 25: quercetin 3-O-rhamnoside; 26: kaempferol 3-O-arabinopyranoside; 27: kaempferol 3-O-arabinofuranoside; 28: kaempferol 3-O-rhamnoside; 29: cyanidin 3-O-galactoside; 30: cyanidin 3-O-glucoside; 31: cyanidin 3-O-arabinoside; 32: peonidin 3-O-galactoside; 33: cyanidin 7-O-arabinoside; 34: cyanidin 3-O-xylosid.

**Figure 2 foods-13-03465-f002:**
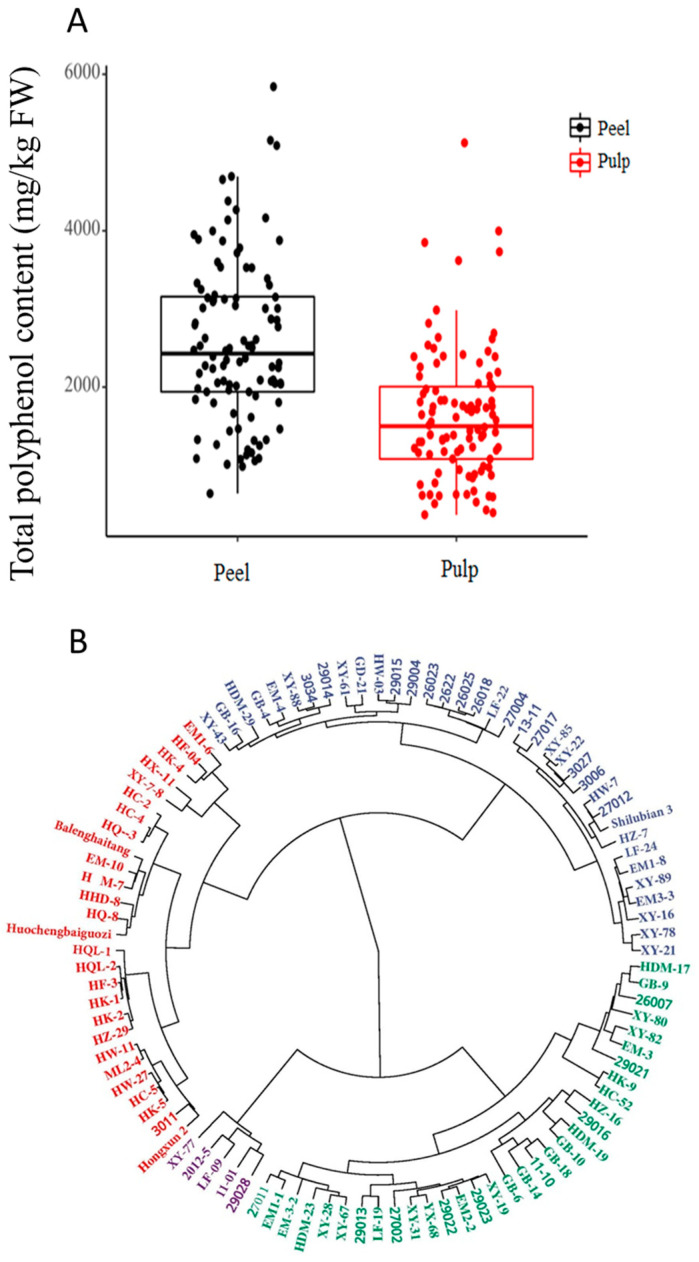
Box diagram of total polyphenolic content in peel and pulp of *Malus* germplasm resources (**A**) and cluster diagram of 100 *Malus* germplasm resources (**B**).

**Figure 3 foods-13-03465-f003:**
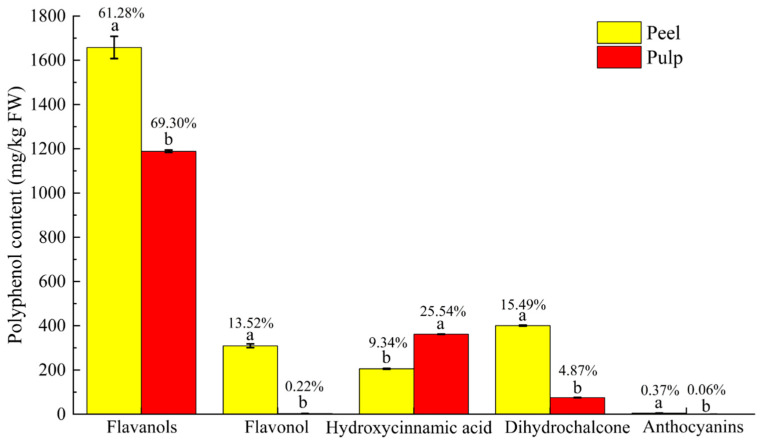
Average content and proportion of five types of polyphenols in peel and pulp for total polyphenolic content. Note: Different letters indicate differences at *p* < 0.05 levels.

**Figure 4 foods-13-03465-f004:**
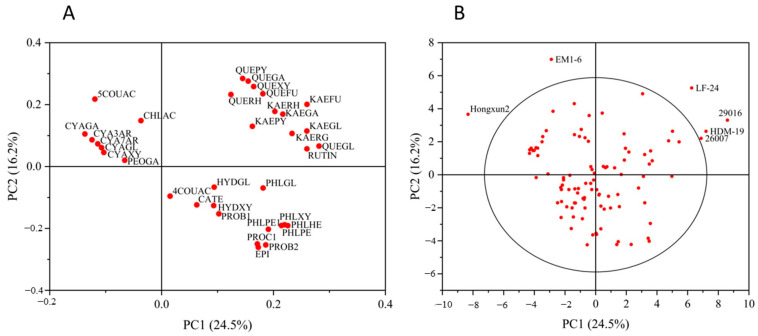
Load plot (**A**) and score plot (**B**) of principal component analysis of peel polyphenol components in *Malus* germplasm resources. Note: ◯ represents a 95% confidence interval.

**Figure 5 foods-13-03465-f005:**
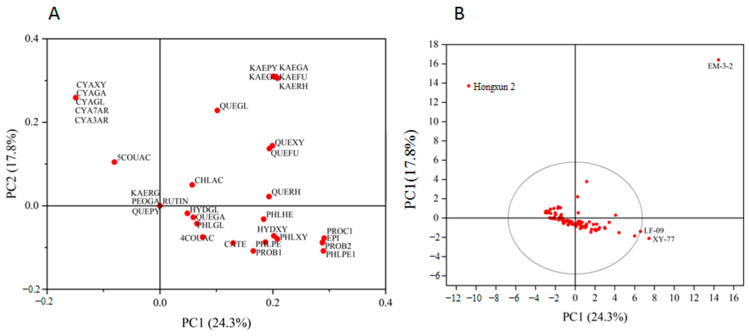
Load plot (**A**) and score plot (**B**) of principal component analysis of pulp polyphenol components in *Malus* germplasm resources. Note: ◯ represents a 95% confidence interval.

**Figure 6 foods-13-03465-f006:**
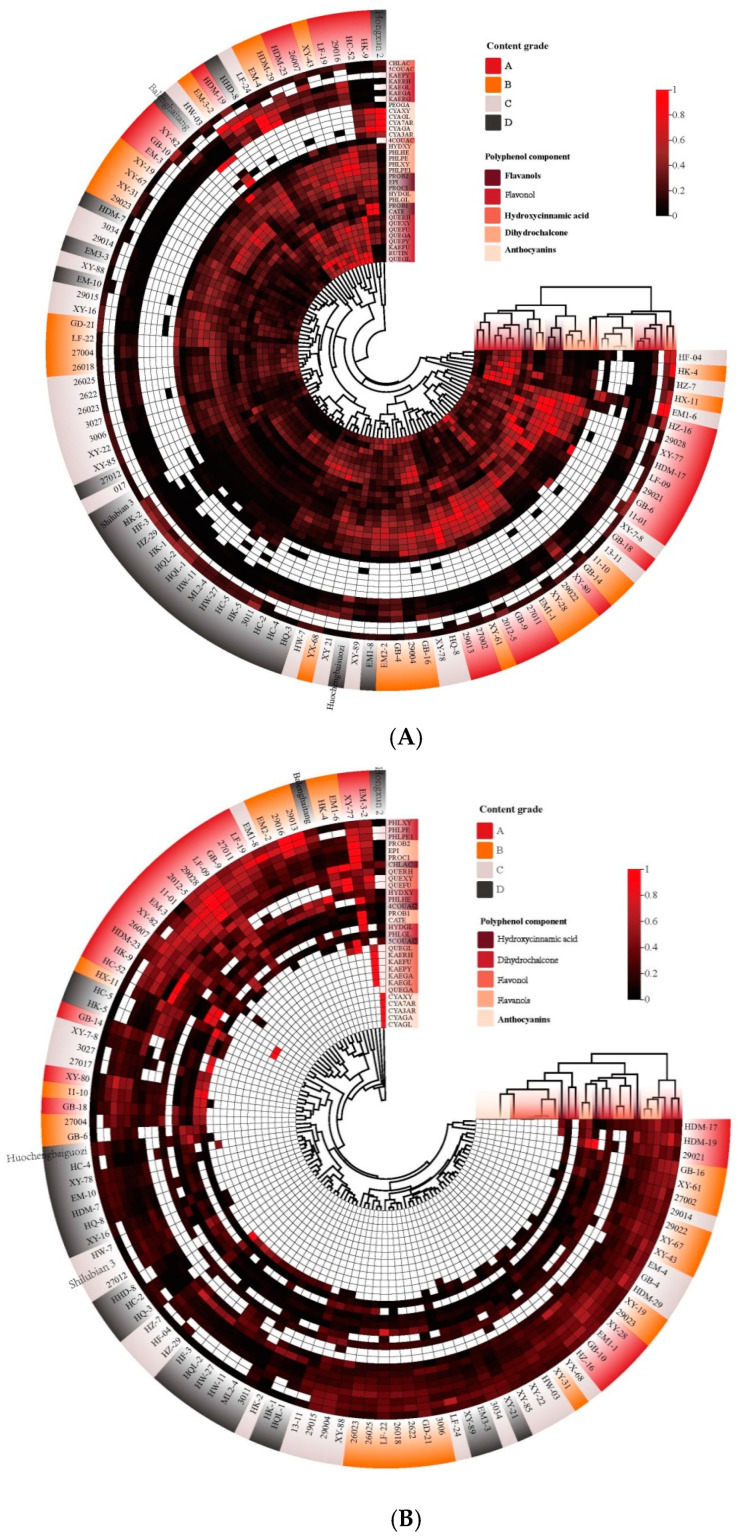
Cluster heatmap of polyphenol content in the peel (**A**) and pulp (**B**) of *Malus* germplasm resources at four levels.

**Table 1 foods-13-03465-t001:** The average content of total polyphenols in the peel and pulp of *Malus* germplasm from different types.

Fruit Part	Type	Average Total Polyphenol Content (mg/kg FW)
Peel	Wild resources	2842.28
	Local variety	1744.36
Pulp	Wild resources	1809.63
	Local variety	1058.25

**Table 2 foods-13-03465-t002:** The average contents and proportions of polyphenol components of *Malus* germplasm resources.

Polyphenol Component	Peel (mg·kg^−1^ FW)	Proportion (%)	Pulp (mg·kg^−1^ FW)	Proportion (%)
PROB1	87.82	3.41%	57.65	3.54%
CATE	47.14	1.83%	37.07	2.28%
PROB2	607.24	23.55%	456.95	28.05%
EPI	486.89	18.88%	364.78	22.39%
PROC1	428.63	16.62%	272.16	16.70%
RUTIN	12.33	0.48%	nd	nd
QUEGA	85.45	3.31%	0.1065	0.01%
QUEGL	60.76	2.36%	0.065	0.00%
QUEXY	31.17	1.21%	0.381	0.02%
QUEPY	4.43	0.17%	nd	nd
QUEFU	67.44	2.62%	0.4151	0.03%
QUERH	42.98	1.67%	2.4229	0.15%
KAEGA	0.7488	0.03%	0.0019	0.00%
KAEGL	1.9377	0.08%	0.0029	0.00%
KAERG	0.6156	0.02%	nd	nd
KAEPY	0.0105	0.00%	0.0013	0.00%
KAEFU	1.0937	0.04%	0.0147	0.00%
KAERH	0.513	0.02%	0.0089	0.00%
CHLAC	178.92	6.94%	311.32	19.11%
4COUAC	24.57	0.95%	48.64	2.99%
5COUAC	1.82	0.07%	1.9	0.12%
HYDXY	4.56	0.18%	4.42	0.27%
PHLHE	2.74	0.11%	1.1431	0.07%
HYDGL	7.55	0.29%	0.7567	0.05%
PHLXY	205.84	7.98%	34.26	2.10%
PHLPE	8.72	0.34%	2.75	0.17%
PHLPE1	6.59	0.26%	5.02	0.31%
PHLZI	164.93	6.40%	26.76	1.64%
CYAGA	4.59	0.18%	0.2665	0.02%
CYAGL	0.0126	0.00%	0.0021	0.00%
CYA3AR	0.1217	0.00%	0.0034	0.00%
PEOGA	0.328	0.01%	nd	nd
CYA7AR	0.1022	0.00%	0.0107	0.00%
CYAXY	0.1628	0.01%	0.0279	0.00%

Note: nd: No detected.

## Data Availability

The original contributions presented in the study are included in the article/Appendix A, further inquiries can be directed to the corresponding author.

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
