# Peer review of "Identification and Mining of Functional Components of Polyphenols in Fruits of Malus Germplasm Resources Based on Multivariate Analysis"

_foods, 2024, doi:10.3390/foods13213465_

Round 1
Reviewer 1 Report
Comments and Suggestions for Authors
Wang et al reported in the manuscript the chemical analysis of polyphenols in several fruits of Malus germplasm resource. The paper needs some clarification about the experimental details and the results section. In addition, an extensive language and typing errors correction should be necessary. In conclusion, I believe that this manuscript could represent a valid innovation of great interest, but a major revision is required for the manuscript.
Revisions:
- Material and methods: Please, add a separate paragraph with the description of the HPLC-DAD-MS method (instrument, chromatographic method, phases, column, flow rate, column temperature, approach of compounds identification and quantification, mass spectrometry parameters, analytical validation of the method, precision, accuracy, linearity ranges, limits of detection (LODs), limits of quantification (LOQs)). In addition, it should be better to specify how the compounds identification was performed (e.g., comparison with analytical standard with the co-injection techniques, analysis of the mass fragmentation pattern)
- Results, subparagraph 3.1, lines 129-131: Please, check some typing errors and spaces in the list of flavanols and dihydrochalcones.
- Results, subparagraph 3.1, line 134: Please, specify how the identification of 4-COAUC and 5-COAUC was performed (comparison with analytical standard with the co-injection techniques, analysis of the mass fragmentation pattern). Due to their regioisomerism, these compounds are characterized by some differences in the mass fragmentation pattern.
- Results, subparagraph 3.1: It should be better to add a table of mass spectrometry characterization of polyphenols, containing the retention times, the m/z, the type of recognized adduct, the mass fragmentation, the literature reference of comparison.
- Results, subparagraph 3.2, lines 176-178: The authors described the statistical approach used for the cluster analysis. It should be better to add these details in the statistical section in Materials and Methods.
- Abstract, line 33 and Conclusion, line 492: The authors reported that the high polyphenol content of the apples analyzed in the study could be used for the development of functional fruit juice. It should be better to comment polyphenols biological activities in Discussion section to support this conclusive consideration.
- General comment: It should be better to make a typing errors correction of the entire manuscript, by checking spaces and some words in the authors native language (e.g. Table 5).
Comments on the Quality of English LanguageAn extensive language and typing errors correction should be necessary.
Author Response
|
Response to Reviewer 1 Comments
|
||
|
1. Summary |
|
|
|
Thank you very much for taking the time to review this manuscript. Please find the detailed responses below and the corresponding revisions/corrections highlighted/in track changes in the re-submitted files.
|
||
|
2. Questions for General Evaluation |
Reviewer’s Evaluation |
Response and Revisions |
|
Does the introduction provide sufficient background and include all relevant references? |
Can be improved |
[Thank you for your suggestions. We already improved the introduction] |
|
Is the research design appropriate? |
Can be improved |
[Thank you for your suggestions. We already added the design details] |
|
Are the methods adequately described? |
Must be improved |
[Thank you for your suggestions. We already improved the methods] |
|
Are the results clearly presented? |
Must be improved |
[Thank you for your suggestions. We already improved the resulsts] |
|
Are the conclusions supported by the results? |
Can be improved |
[Thank you for your suggestions. We already improved the conclusion] |
|
3. Point-by-point response to Comments and Suggestions for Authors |
||
|
Comments 1: Material and methods:Please, add a separate paragraph with the description of the HPLC-DAD-MS method (instrument, chromatographic method, phases, column, flow rate, column temperature, approach of compounds identification and quantification, mass spectrometry parameters, analytical validation of the method, precision, accuracy, linearity ranges, limits of detection (LODs), limits of quantification (LOQs)). In addition, it should be better to specify how the compounds identification was performed (e.g., comparison with analytical standard with the co-injection techniques, analysis of the mass fragmentation pattern) |
||
|
Response 1: Thank you for pointing this out. We have, accordingly, added the relevant content and specific steps in the ‘Methods’. |
||
|
Comments 2: Results, subparagraph 3.1, lines 129-131: Please, check some typing errors and spaces in the list of flavanols and dihydrochalcones. |
||
|
Response 2: Thank you for pointing this out. We have, accordingly, revised the typing errors and spaces in the whole manuscript. |
||
|
Comments 3: Results, subparagraph 3.1, line 134: Please, specify how the identification of 4-COAUC and 5-COAUC was performed (comparison with analytical standard with the co-injection techniques, analysis of the mass fragmentation pattern). Due to their regioisomerism, these compounds are characterized by some differences in the mass fragmentation pattern. |
||
|
Response 3: Thank you for pointing this out. We have, accordingly, add the specific methods and reference. |
||
|
Comments 4: Results, subparagraph 3.1: It should be better to add a table of mass spectrometry characterization of polyphenols, containing the retention times, the m/z, the type of recognized adduct, the mass fragmentation, the literature reference of comparison. |
||
|
Response 4: Thank you for pointing this out. Our laboratory has been using this mehtod for the last few years, it is an effenitve and stable method, and we have published and cited relevant references. We have, accordingly, add the specific methods and reference. Therefore, these parameters and tables are not listed in the manuscript. |
||
|
Comments 5: Results, subparagraph 3.2, lines 176-178: The authors described the statistical approach used for the cluster analysis. It should be better to add these details in the statistical section in Materials and Methods. |
||
|
Response 5: Thank you for pointing this out. We have, accordingly, these details in the statistical section in ‘Methods’. |
||
|
Comments 6: Abstract, line 33 and Conclusion, line 492: The authors reported that the high polyphenol content of the apples analyzed in the study could be used for the development of functional fruit juice. It should be better to comment polyphenols biological activities in Discussion section to support this conclusive consideration. |
||
|
Response 6: Thank you for pointing this out. We have, accordingly, added the relevant content and references in the ‘Discussion’. |
||
|
Comments 7: General comment: It should be better to make a typing errors correction of the entire manuscript, by checking spaces and some words in the authors native language (e.g. Table 5). |
||
|
Response 7: Thank you for pointing this out. We have, accordingly, made a typing errors correction of the entire manuscript and made English edit using MDPI Language Editing Services. |
||
|
|
||
|
|
||
Thank you so much again. We will revise it carefully again, if further comments are suggested.

Reviewer 2 Report
Comments and Suggestions for Authors
Wang et al. have aimed to determine the polyphenol composition in peel and pulp of 74 Malus sieversii and 26 local varieties grown in apple germplasm resources by UPLC and LC-MS as well as performed multivariate analysis of content data by principal component analysis and hierarchical clustering analysis. This study contains some important analytical data which should be of interest in the field. However, the following points need to be addressed:
1. The abstract should include information on the multivariate analysis performed in the study involving principal component analysis and hierarchical clustering analysis.
2. The multivariate analysis details should also be included in the title and conclusion.
3. The keywords such as ‘evaluation’ and ‘screening’ should be removed. Instead ‘Malus sieversii’, ‘apple fruit’, ‘Rosaceae’, ‘multivariate analysis’, ‘UPLC polyphenol analysis’, ‘LC-MS’ should be included as more relevant keywords aligning with the content of this article.
4. At the end of introduction, precise objectives should be written in one or two sentences with clarity.
5. Tables 1, 4, 5, 6, and 7 should be moved to the supplementary materials and only the important details discussed in the text.
6. Why was the proximate analysis of some representative samples not performed in this study?
7. The section 2 is missing the UPLC and LC-MS methods used for determination of content of polyphenols.
8. The information of replicates is missing in section 2.3.
9. What are chromatograms in Figure 1A, B, C and D represent? They should be specified in the Figure 1 captions.
10. The total phenol content plotted in Figure 2 corresponds to the summation of individual polyphenols as determined by UPLC and LC-MS or by determination of total phenolic content directly.
11. Figures 2 and 3 should be combined as A and B parts of one figure.
12. Figures 7 and 8 should be combined as A and B parts of one figure.
13. Some future perspectives should be included at the end of conclusion.
Author Response
|
Response to Reviewer 2 Comments
|
||
|
1. Summary |
|
|
|
Thank you very much for taking the time to review this manuscript. Please find the detailed responses below and the corresponding revisions/corrections highlighted/in track changes in the re-submitted files.
|
||
|
2. Questions for General Evaluation |
Reviewer’s Evaluation |
Response and Revisions |
|
Does the introduction provide sufficient background and include all relevant references? |
Yes |
|
|
Is the research design appropriate? |
Must be improved |
[Thank you for your suggestions. We already added the design details] |
|
Are the methods adequately described? |
Must be improved |
[Thank you for your suggestions. We already improved the methods] |
|
Are the results clearly presented? |
Must be improved |
[Thank you for your suggestions. We already improved the results] |
|
Are the conclusions supported by the results? |
Can be improved |
[Thank you for your suggestions. We already improved the conclusion] |
|
3. Point-by-point response to Comments and Suggestions for Authors |
||
|
Comments 1: The abstract should include information on the multivariate analysis performed in the study involving principal component analysis and hierarchical clustering analysis. |
||
|
Response 1: Thank you for pointing this out. We have, accordingly, added relevant content in the abstract. |
||
|
Comments 2: The multivariate analysis details should also be included in the title and conclusion. |
||
|
Response 2: Thank you for pointing this out. We have, accordingly, added relevant content in the title and conclusion. |
||
|
Comments 3: The keywords such as ‘evaluation’ and ‘screening’ should be removed. Instead ‘Malus sieversii’, ‘apple fruit’, ‘Rosaceae’, ‘multivariate analysis’, ‘UPLC polyphenol analysis’, ‘LC-MS’ should be included as more relevant keywords aligning with the content of this article. |
||
|
Response 3: Thank you for pointing this out. We have, accordingly, removed the old keywords and added the new keywords suggested. |
||
|
|
||
|
Comments 4: At the end of introduction, precise objectives should be written in one or two sentences with clarity. |
||
|
Response 4: Thank you for pointing this out. We have, accordingly, added relevant content at the end of introduction. |
||
|
Comments 5: Tables 1, 4, 5, 6, and 7 should be moved to the supplementary materials and only the important details discussed in the text. |
||
|
Response 5: Thank you for pointing this out. We have, accordingly, removed the Tables 1, 4, 5, 6, and 7 to supplementary as Table S1, S2, S3, S4, and S5. |
||
|
Comments 6: Why was the proximate analysis of some representative samples not performed in this study? |
||
|
Response 6: Thank you for pointing this out. PCA and HCA has been carried out in the manuscript, therefore no approximate anaylysis was performed. |
||
|
Comments 7: The section 2 is missing the UPLC and LC-MS methods used for determination of content of polyphenols. |
||
|
Response 7: Thank you for pointing this out. We have, accordingly, added relevant content in the ‘Methods’. |
||
|
Comments 8: The information of replicates is missing in section 2.3. |
||
|
Response 8: Thank you for pointing this out. There is the replicate information in section 2.1. |
||
|
Comments 9: What are chromatograms in Figure 1A, B, C and D represent? They should be specified in the Figure 1 captions. |
||
|
Response 9: Thank you for pointing this out. We have, accordingly, added title for the Figure1 A, B, C, D, respectively. |
||
|
Comments 10: The total phenol content plotted in Figure 2 corresponds to the summation of individual polyphenols as determined by UPLC and LC-MS or by determination of total phenolic content directly. |
||
|
Response 10: Thank you for pointing this out. We have, accordingly, added relevant content in the ‘Methods’, the total polyphenol content was the summation of the individual polyphenol components. |
||
|
Comments 11: Figures 2 and 3 should be combined as A and B parts of one figure. |
||
|
Response 11: Thank you for pointing this out. We have, accordingly, combined Figure 2 and Figure 3 to A and B of Figure 2.
Comments 12: Figures 7 and 8 should be combined as A and B parts of one figure. |
||
|
Response 12: Thank you for pointing this out. We have, accordingly, combined Figure 7 and Figure 8 to A and B of Figure 6.
Comments 13: Some future perspectives should be included at the end of conclusion. |
||
|
Response 13: Thank you for pointing this out. We have, accordingly, added future perspectives in the conclusion.
|
||
Thank you so much again. We will revise it carefully again, if further comments are suggested.

Round 2
Reviewer 2 Report
Comments and Suggestions for Authors
The authors have satisfactorily addressed all the comments raised by reviewers and substantially improved the overall quality of the article. Therefore, I recommend accepting this article for publication in Foods.